# Residents’ Awareness of Family Doctor Contract Services, Status of Contract with a Family Doctor, and Contract Service Needs in Zhejiang Province, China: A Cross-Sectional Study

**DOI:** 10.3390/ijerph16183312

**Published:** 2019-09-09

**Authors:** Xiaopeng Shang, Yangmei Huang, Bi’e Li, Qing Yang, Yanrong Zhao, Wei Wang, Yang Liu, Junfen Lin, Chonggao Hu, Yinwei Qiu

**Affiliations:** 1Zhejiang Provincial Center for Disease Control and Prevention, Hangzhou 310051, China (X.S.) (Q.Y.) (Y.Z.) (W.W.) (J.L.); 2The Center for Disease Control and Prevention of Hangzhou City, Hangzhou 310021, China; 3Danzhou City Center for Disease Control and Prevention, Danzhou 571700, China; 4School of Public Health, Xiamen University, Xiamen 361102, China

**Keywords:** family doctor contract services, awareness, demands, health care reform, Zhejiang Province

## Abstract

In China, family doctor services originated in 2009. After two years, the Chinese government proposed the establishment of a family doctor contract system suitable for China’s national conditions. Then, in 2016, a multi-department jointly issued an important document, which further clarified the development goals of family doctor contract services in the next five years. Zhejiang Province has been exploring responsible doctor contract services since 2012, which was promoted throughout the province in 2015. Objectives: The aim of this study was to investigate the residents’ awareness of Zhejiang Province, China, of family doctor contract services, the status of signing such a contract, and the demand for service items in the contracted service package. Further, we sought to explore the relevant influential factors in order to provide a reference and evidence-based recommendations for the further development of family doctor contract services. Design: We enrolled 3960 residents from nine counties in Zhejiang Province using a multistage stratified random sampling method. A survey using a self-designed questionnaire was used to collect the demographic data, residents’ awareness of family doctor contract services, the status of contracting, and demand for different items from October to December 2017. Data were analyzed by SPSS 21.0. Results: In total, 3871 residents returned valid questionnaires, with a response rate of 97.75%. The awareness rate of residents of family doctor contract services was 71.58% (2771/3871). Age, education level, and chronic medical history status were the influencing factors affecting residents’ awareness. The contracted rate was 50.43% (1952/3871). Age, education level, personal monthly income, chronic disease history, and awareness of family doctor contract services were the influencing factors. Residents who have a contract with family doctors have a higher demand for family doctor contract services, and different residents have different needs for the project because of their physical condition, education level, marital status, household registration, and personal monthly income level. The top three needs of the residents for contracted services were health consultation (84.64%), regular physical examination (81.71%), and increasing the proportion of medical insurance reimbursements (80.06%). Conclusions: The awareness rate of family doctor contract services and the contracting rate are unsatisfactory among residents of Zhejiang Province. It is suggested that the government should more heavily publicize family doctor contract services, expand the coverage, introduce personalized contract schemes to meet the needs of different groups, and promote the rapid development of family doctor contract services in Zhejiang Province.

## 1. Introduction

In the past decade, the demand for medical services among Chinese residents has steadily grown. Some studies have shown that the number of medical treatments in primary health care institutions has increased, but the proportion of medical treatments has declined, which has become a major obstacle for the promotion of China’s hierarchical medical system [1]. The family doctor (FD) system in China, also known as the general practitioner (GP) or the family physician (FP) system, is ultimately intended to promote the establishment of primary care, bidirectional referral mechanisms between primary care centers and secondary or tertiary hospitals, a hierarchical medical system, and the mechanism of division of labor and cooperation between public hospitals and primary health care institutions [2,3]. Family doctor contract services (FDCS) are essentially the extension and development of community health services based on the principle of full notification, voluntary signing, and standardized service, providing a proactive, continuous, and comprehensive health accountability management model by signing service contracts with residents. In early 2009, the Central Committee of the Communist Party of China and the State Council issued the “Opinions on Deepening the Reform of the Medical and Health System” [4], which emphasized providing active, continuous, and accountable services and comprehensively developing various models of family doctor services. At the same time, it also inaugurated the prelude to China’s new health care reform [5]. In 2011, the State Council issued the “Guiding Opinions on Establishing a General Practitioner System” [6], which proposed the establishment of a general practitioner system suitable for China’s national conditions, to establish a hierarchical medical system, and implement a GP service. In May 2016, the State Council’s Medical Reform Office, the National Health Commission, the National Development and Reform Commission, and seven other departments jointly issued the “Guiding Opinions on Promoting Family Doctor Contract Services” [7], which proposed the three-stage goal of launching FDCS in China. In 2016, 200 pilot cities for the comprehensive reform of public hospitals carried out FDCS and other qualified regions were encouraged to actively carry out pilot projects. In 2017, the coverage rate of FDCS reached over 30%, and the coverage rate of key population contracted services reached over 60%. By 2020, the aim is to expand contracted services to the entire population, form a long-term stable contractual service relationship, and basically achieve the goal of full coverage of the FDCS system.

Zhejiang Province has been exploring responsible doctor contract services since 2012, which was promoted throughout the province in 2015 [8,9]. In order to avoid the one-sided pursuit of a high contract rate and neglect the quality of contract services, the Zhejiang Provincial Health Commission issued the “Notice on further implementation of contract services for family doctors” in April 2018 [10]. This document first clarified the unified name of the “family doctor’s contract services” and, at the same time, required all localities to maintain contracted service coverage at 35% and the signing coverage rate of 10 key groups should reach over 65%, based on existing contract services combined with the service capacity and resource allocation. All localities should shift the focus of work to improve quality and efficiency, place the quality of contracted services first, and continuously improve the sense of fulfillment and satisfaction of contracted residents. In addition, a reference list of the contents of family doctor contract services in Zhejiang Province was provided. From the statistical data in the National Primary Health Information System, by the end of 2018, 11 cities and 89 counties in Zhejiang Province had issued contract service policy documents, with a total of 18.44 million contracted residents and a contract rate among all residents of 34.90%. A total of 14.29 million key populations (including the elderly; pregnant women; children; the disabled; special family members with family planning; rural people with difficulties; patients with chronic diseases such as hypertension, diabetes, and tuberculosis; and patients with serious mental disorders) have signed up for FDCS, with a contract rate of 75.18%. There are 1484 primary health care institutions that carry out contracted services, including 493 community health service centers and 991 township health centers, covering 52.83 million permanent residents. The average primary medical institution services 35,596 residents. The number of qualified doctors in Zhejiang Province has reached 24,228, and there are 19,896 registered as GPs—of which 19,341 are GPs who provide contract services and 9603 are GPs who have intermediate professional titles or above. However, the actual signing rate of residents in Zhejiang Province and the extent of the residents’ demand for the contents of the contracted service list are not known, and there are few relevant studies. So, it is imperative to conduct on-the-spot investigations of the signing status of residents and the demand for contracted services, which is significant for demand-oriented health policy research.

This study selected urban and rural residents from nine counties (cities and districts) as the research participants, with the purpose of investigating and understanding the reality of FDCS among residents of Zhejiang Province. We sought to answer the following questions: (1) Are residents aware of the FDCS, and what are the relevant factors affecting residents’ awareness? (2) What is the real contract rate of residents, and what are the factors that affect the registration of residents for FDCS? (3) What are the different needs of residents with different characteristics for the 10 items provided in the FDCS? The solution to these problems will provide recommendations for further improving the content of FDCS, developing personalized and targeted contract services for FDs, and formulating and optimizing relevant policies.

## 2. Materials and Methods

### 2.1. Research Participants

A household-based cross-sectional study was conducted using a face-to-face questionnaire survey in Zhejiang Province. To ensure the demographic representation, the sample quantity was calculated using the formula $n=\frac{\mu_{\alpha }^{2}p\left(1-p\right)}{\delta^{2}}$, with the sample size *n* = 3585. Specifically, confidence level *α* = 0.05 (two sides), $\mu_{\alpha }$ = 1.96, contracted rate *p* = 30% (according to the second-stage goal in the document mentioned above), and allowable error *δ* = 0.015. To facilitate the allocation of places, the integer 3600 was taken as our sample size. With the no-response rate controlled within 10%, the actual sample size was 3960 residents.

Multistage sampling was conducted to select the participants following the three steps below. Firstly, all 90 counties in Zhejiang Province were divided into three levels (good, medium, and poor) based on the GDP ranking in 2016. Three counties were selected randomly by a random number table from each level. Two neighborhood committees and two townships were randomly selected in each county. Secondly, one community/natural village was selected randomly from each neighborhood committee/township. Thirdly, using a system sampling method, 110 households were selected from the household registration list provided by the community or natural village, and one resident was selected from each household, usually the head of the household, making up the total sample of 3960 residents (Figure 1). The eligibility criteria were (1) age (≥16 years), (2) having lived in the district for over six months, and (3) be willing to participate in the survey. The exclusion criterion was those with language or communication barriers. A total of 3960 questionnaires were sent out, while 89 unqualified samples were excluded. Finally, a total of 3871 valid questionnaires were included in the database, with a collection rate of 97.75%.

### 2.2. Methods and Data Collection

Based on the literature and expert consultation, the self-designed questionnaire was compiled according to the purpose of the survey. The survey items mainly covered three parts: (1) demographic information and sociological status, including age, sex, nationality, household registration, education level, marital status, personal income per month, and chronic disease history (hypertension, diabetes, cardiovascular and cerebrovascular diseases, tumors, etc.); (2) residents’ awareness of FDCS and status of signing a contract; (3) the 10 service items selected from the list of the contents of the FDCS [10], including health consultation, follow-up of chronic patients, long-term prescriptions for chronic patients, rehabilitation guidance, appointment referral, regular physical examination, increase in the proportion of medical insurance (MI) reimbursement, Traditional Chinese Medicine (TCM) health care, family bed service, and free door-to-door service to understand the needs of the residents for each item.

From October to December 2017, this survey was conducted in Zhejiang Province. Each county selected four to five investigators from the local Center for Disease Control and Prevention (CDC) and Basic Public Health Service Project Management Office. The criteria for investigators were to speak local dialects, to be good at communicating with others, to have relevant work experience, and to be responsible. Beforehand, they were trained to better understand the survey questionnaire and the skills of face-to-face interviewing. At the beginning of the investigation, the selected households were notified in advance to concentrate in one place to conduct the investigation. For the absent people, the investigators directly conducted the household survey. A small gift was sent to each respondent to improve the residents’ cooperation and response rate. If the selected household was not available, a nearby neighbor was chosen as a replacement, but the replacement rate was strictly controlled within 5%.

### 2.3. Statistical Analysis

A descriptive analysis was conducted to describe the sample characteristics. The categorical variables were calculated by frequency and percentage. A comparison of sociodemographic characteristics between aware and unaware residents, contracted and non-contracted residents, and residents with a need and no need for specific service items was conducted using the Pearson chi-squared test (χ^2^ test). Binary logistic regression was then performed to determine the significant factors that influenced residents’ awareness of FDCS and contracting with FD, using the forward stepwise selection (likelihood ratio) method, taking α = 0.05 as the inclusion index and α = 0.1 as the elimination index [11,12]. The odds ratio (OR) and 95% confidence interval (CI) of the variables were reported.

EpiData version 3.1 software (EpiData Association, Odense, Denmark, Europe) was adopted to establish the database, and the double para entry rule was applied for inputting data. All the analyses were carried out using IBM SPSS V.21.0 (IBM, Chicago, IL, USA), and the threshold of statistical significance was set at *p* < 0.05 (two-tailed).

### 2.4. Ethical Considerations

This study was approved by the academic ethics committee of the Zhejiang Provincial Center for Disease Control and Prevention. Our survey was voluntary, and residents could refuse to participate. All were assured of the confidentiality of their information discussed during the interview.

## 3. Results

### 3.1. Sociodemographic Information of Participants

The sociodemographic data of the participants are listed in Table 1 with the number (*N*) and the corresponding percentages. In total, 3871 participants were included in our analysis—of whom, 1571 (40.58%) were male and 2300 (59.42%) were female, with a sex ratio of 1:1.46. The mean age was 51.05 (SD = 16.86) years. Age was divided into four groups: 16–34, 35–49, 50–64, and 65 years or more. The largest proportion of participants (32.03%) was in the 65 years or more group, and the second largest proportion (26.87%) was in the 30–49 years age group. Most of the respondents had an education level of elementary school or lower (38.41%), followed by junior middle school (26.69%). The number of married persons was 3249 (83.93%). In terms of household registration, 97.78% were registered as local. As for monthly income, 84.91% belonged to the CN¥0–5000 group. Additionally, 39.16% of respondents had chronic disease history.

### 3.2. Residents’ Awareness of FDCS and Influencing Factors

Of the 3871 respondents, 2771 stated explicitly that they were aware of family doctor contract services, with an awareness rate of 71.58% (Table 1). The Pearson chi-squared test method was used to assess the correlation between all indexes of sociodemographic factors and awareness of FDCS. The results showed that age, education level, personal income (per month), and chronic disease history were significantly related to residents’ awareness of FDCS (*p* < 0.05).

Binary logistic regression was used to judge the factors significantly associated with the residents’ awareness of FDCS. Table 2 shows the relationship between individual factors and residents’ awareness. Age, education level, and chronic disease history were the sociodemographic factors that had a significant impact on residents’ awareness. To be specific, older residents were more likely to understand FDCS than younger residents (OR = 1.567, 95% CI: 1.262–1.969, *p* < 0.001; OR = 1.935, 95% CI: 1.481–2.528, *p* < 0.001; OR = 2.781, 95% CI: 2.098–3.688, *p* < 0.001). Residents with a higher education level were more likely to be aware of FDCS than those with a lower level (OR = 1.302, 95% CI: 1.066–1.590, *p* = 0.010; OR = 1.614, 95% CI: 1.259–2.069, *p* < 0.001; OR = 3.025, 95% CI: 2.181–4.196, *p* < 0.001; OR = 4.623, 95% CI: 3.347–6.386, *p* < 0.001). Residents who had a chronic disease history were 1.058 times more likely to be aware of FDCS than those who did not have a chronic disease history (OR = 2.880, 95% CI: 2.390–3.472, *p* < 0.001).

### 3.3. Residents Who Had a Contract with FDs and Influencing Factors

As illustrated in Table 1, there were 1952 residents who had a contract with their family doctors, with a contract rate of 50.43%. Age, education level, marital status, household registration, personal income (per month), and chronic disease history were significantly related to residents contracting with FDs (*p* < 0.05).

The logistic regression model used the contracted status (contracted and non-contracted) as the dependent variable, the factors significantly associated with contracting with FDs (age, education level, marital status, household registration, personal income, and chronic disease history), and residents’ awareness of FDCS or not as independent variables. Table 3 shows that age, education level, personal income (per month), chronic disease history, and awareness of FDCS were the factors that had a significant impact on residents contracting with FDs. Residents of the age groups 50–64 and ≥65 years were more likely to have a contract with FDs than those of the 16–34 years age group (OR = 1.474, 95% CI: 1.096–1.982, *p* = 0.01; OR = 2.656, 95% CI: 1.959–3.601, *p* < 0.001). Residents with a bachelor’s degree or higher were less likely to have a contract with FDs than those with an educational background of elementary school or lower (OR = 0.586, 95% CI: 0.417–0.825, *p* = 0.002). Compared with residents with a monthly income of ≤CN¥2000, residents with a monthly income of CN¥2001–5000 were less inclined to have a contract with FDs (OR = 0.739, 95% CI: 0.610–0.894, *p* = 0.002), and residents with a monthly income of CN¥8000 were more likely to have a contract with FDs (OR = 1.829, 95% CI: 1.198–2.791, *p* = 0.005). Residents who had a chronic disease history were 2.629 times more likely to have a contract with FDs than those who did not have a chronic disease history (OR = 2.629, 95% CI: 2.170–3.186, *p* < 0.001). Residents who were aware of FDCS were 22.753 times more likely to have a contract with FDs than those who were unaware (OR = 22.753, 95% CI: 17.870–28.971, *p* < 0.001).

### 3.4. Residents’ Need for the 10 Items of FDCS

Table 4 shows that the respondents’ need for the 10 different items in the contracted services with the number and the corresponding percentages, as well as the results of correlation between all indexes of sociodemographic factors and residents’ needs with chi-squared values and *p*-values. The top three needs of the residents for contracted services were health consultation (84.64%), regular physical examination (81.71%), and increasing the proportion of medical insurance reimbursements (80.06%).

More women than men were in demand for follow-up of chronic patients and free physical examinations (*p* < 0.05). Age was correlated with nine other items in contract services except TCM health care (*p* < 0.01). Also, there appeared to be a tendency for the demand for almost all items to increase with age. Education level was related to follow-up of chronic patients, appointment referral, free physical examinations, TCM health care, and family sickbeds (*p* < 0.001). Marital status was correlated with health consultation, follow-up of chronic patients, long-term prescriptions for chronic patients, free physical examinations, and increasing the proportion of medical insurance reimbursements (*p* < 0.01). Local residents had a higher demand for follow-up of chronic patients than non-local residents (*p* < 0.05). Personal monthly income was related to follow-up of chronic patients, appointment referral, free physical examinations, increasing the proportion of medical insurance reimbursements, and TCM health care. Residents who have a chronic disease history have a greater need for health consultation, follow-up of chronic patients, long-term prescriptions for chronic patients, rehabilitation guidance, free physical examinations, and free door-to-door services than those who do not have a chronic disease history (*p* < 0.001). For all 10 items, contracted residents have more need than those who do not have contracts (*p* < 0.01).

## 4. Discussion

It is a key task for China to deepen the reform of the medical and health care system by fulfilling demand-oriented contract services with family doctors [13]. This requires fully understanding the real needs of residents for family doctor contract services before formulating health policies.

Our research showed that the awareness rate of residents for FDCS was 71.58% in Zhejiang in 2017, which was lower than that in Beijing in 2014 (84.4%) [14] and Guangzhou in 2016 (81.53%) [15]. This rate is, however, slightly higher than those of residents surveyed in the Shanghai Minhang District (65.8%) [16] and the Changning District (67.1%) in 2016 [17]. The reason for this discrepancy is that, on the one hand, Beijing, Guangdong, and Shanghai are all pilot areas, where FDCS were implemented earlier than in Zhejiang [18]. On the other hand, this also shows that there is still much room for improving the awareness rate of FDCS for residents in Zhejiang Province. The multielement stepwise regression analysis showed that the influencing factors included age, education level, and chronic disease history. Specifically, residents who were older, had a higher education level, and had chronic diseases had a higher awareness rate of contracted services. It is not difficult to imagine that the elderly and residents with chronic diseases are more inclined to pay attention to their own health and the acquisition of health knowledge, thus making their awareness rate of FDCS relatively higher. People with high levels of knowledge have more access to knowledge and a better understanding of the country’s latest policies, so their awareness of FDCS is also higher.

In the current survey, the contract rate in Zhejiang Province in 2017 was 50.43%, which was lower than that in the Shanghai Hongkou District in 2015 (65.3%) [19], the Shanghai Changning District in 2014 (52.8%) [20], and Guangzhou Province in 2015 (52.9%) [21]. This is basically consistent with the difference in awareness rate mentioned above. The signing service for FDs in Zhejiang Province starting late was one of the main reasons. According to the implementation target of the second phase of the Zhejiang Provincial Health Commission, the 50.43% signing rate in 2017 also reached the target [7]. In accordance with the binary logistic regression results, we found that contracted residents were older, had a lower education level, had a personal monthly income of ≤CN¥2000 and included fewer people from the higher monthly income group of ≥CN¥8001, had a chronic disease history, and had awareness of FDCS. Similar results were found in previous studies performed in Guangdong Province and Shanghai, which all showed that contracted participants were older, had a higher awareness rate of FDCS, and were less healthy compared with non-contracted participants [2,19,22,23]. The government usually gives priority to the poor, chronic disease patients, the elderly, and the disabled, who have higher demands for community health services. This group of people, as the key group driving the government to implement the FDCS policy, is given priority due to the convenience brought by FDCS, and the program is to be expanded to the entire population gradually [24,25,26,27]. So, current contracted residents mainly include people belonging to such key groups. In addition, for residents, the more familiar they are with FDCS, the higher the utilization rate, which will increase the rate of contracting with FDs [14]. Further, contracting with FDs can also help residents better understand FDCS.

There were 913 participants who knew of family doctor contract services but did not sign up for them. Most of these were young people who had a high level of education, were married, were locals, had a moderate personal income level, and had no chronic disease history. Specifically, among such residents, 563 (61.66%) were female, 595 (65.17%) were younger than 50 years, 301 (32.97%) had an education level of junior college or higher, 757 (82.91%) were married, 888 (97.26%) were local residents, 457 (50.05%) had a personal monthly income of CN¥2001–5000, and 705 (77.21%) had no chronic disease history. For this group of people, we should first understand their needs and then provide corresponding services to meet their needs to attract them to sign up. Also, 94 of those who were not knowledgeable about FDCS had signed a contract. Most of them were elderly who had low levels of education, were married, were locals, had a low monthly income, and no chronic disease history. Specifically, 59 (62.77%) were female, 67 (71.28%) were older than 50 years, 59 (62.77%) had an education level of elementary school or lower, 71 (75.53%) were married, 93 (98.94%) were local residents, 56 (59.57%) had a personal monthly income of ≤2000, and 50 (53.19%) had no chronic disease history. For such residents, it is necessary to strengthen publicity and encourage them to participate in contracted services.

With the expansion of the scope of signing contracts for family doctors, the government should design the contract service content according to the specific demands of different populations in order to further promote the sustainable development of the FDCS system [24]. The current survey showed that different participants have different needs for the project because of their physical condition, education level, marital status, household registration, and personal monthly income level. For the overall demand, the project items with a demand rate higher than 60% are health consultation (84.63%), regular physical examinations (81.71%), increasing the proportion of medical insurance reimbursements (80.06%), follow-up of chronic patients (65.87%), rehabilitation guidance (64.89%), and appointment referral (62.62%). Residents’ high demand for health consultation and regular physical examinations is consistent with the results of similar studies in China [28,29]. Residents’ demand for TCM health care, free door-to-door service, long-term prescriptions for chronic patients, and family bed service is relatively low. This result is inconsistent with the survey results of the Shanghai Changning District in 2013, which showed that rehabilitation guidance, TCM health care, and appointment referral were the top three demands among residents for FDCS [20], and the survey results of the Beijing Xicheng District in 2014, which showed that residents have the highest demand for TCM health care [30]. The reason for these survey results may be that the implementation of TCM health care services for contracted residents in Zhejiang Province started later than in pilot areas such as Beijing and Shanghai, and residents have little understanding of the services. It is also possible that due to different economic development levels and geographical locations, Zhejiang Province residents tend to pay more attention to their own health assessments, physical examinations, and increasing the proportion of medical insurance reimbursements. For people of different genders, women seem to have a higher need for follow-up of chronic diseases and regular physical examinations. As age increases, the demand rate among residents for various services (except TCM health care) also increases. People with a lower education level have a higher demand for follow-up of chronic patients and regular physical examinations, while those with a higher education level have a higher demand for appointment referrals, TCM health care, and family bed service. Divorced or widowed residents are more likely to receive health consultations, follow-up of chronic patients, long-term prescriptions for chronic patients, regular physical examinations, and to want to increase the proportion of MI reimbursements than single or married people. Local residents have a higher demand for follow-up of chronic patients than non-local residents. People with lower monthly income levels have a higher demand for follow-up of chronic patients, regular physical examinations, and increasing the proportion of MI reimbursements, while those with higher monthly incomes have a higher demand for appointment referrals and TCM health care. Residents with a chronic medical history have a high demand rate for all items of contracted services, especially health consultation, follow-up of chronic patients, long-term prescriptions for chronic patients, rehabilitation guidance, and regular physical examinations.

Except for the sociodemographic characteristics, we found that the contracted residents’ demand for all project items was higher than that of non-contracted residents. This is in line with the initial intention of having residents sign contracts with family doctors, which is to obtain the required service items [31]. Our study suggests that residents with different characteristics have different requirements for the projects in the contract service package. In order to better meet the service needs of residents, primary health care institutions should provide targeted services based on the characteristics of the residents.

There are a few limitations of this study. On the one hand, this was a cross-sectional study using a multistage stratified random sampling method to select respondents. As we know, most young people go out to work and the elderly stay at home, which may lead to underrepresentation of the whole population in Zhejiang Province. On the other hand, in the survey of residents’ needs for FDCS, we only screened the 10 most common items in the service package and did not cover all items. This requires us to conduct a more comprehensive investigation in the future.

## 5. Conclusions

Both the awareness level of FDCS and the rate of signing a contract with a family doctor are unsatisfactory among residents in Zhejiang Province, and there is room for further improvement. Age, education level, and chronic medical history are the influencing factors affecting residents’ awareness of FDCS. At the same time, residents’ knowledge of FDCS affects their rate of contracting with family doctors. Residents who have a contract with family doctors have a higher demand for FDCS, but residents with different sociodemographic characteristics have different demand rates for different FDCS items. Therefore, in order to promote the development of FDCS, the government should provide greater policy support, increase publicity, expand service packages, and provide more attractive service projects. Residents can choose a family doctor with whom to sign a contract. When signing the contract, residents can independently choose the items they need. At the same time, family doctors should provide better services to enhance residents’ sense of fulfillment and satisfaction.

## Figures and Tables

**Figure 1 ijerph-16-03312-f001:**
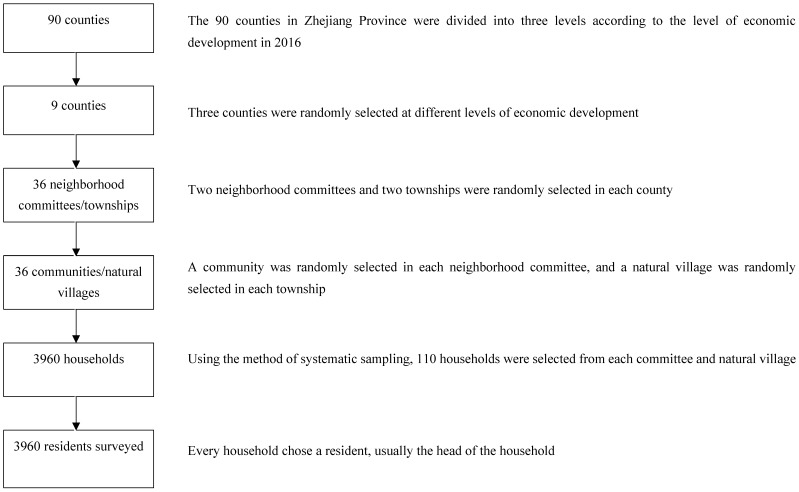
Flowchart of the sampling method.

**Table 1 ijerph-16-03312-t001:** Awareness of family doctor contract services (FDCS) and signing status among residents of Zhejiang Province.

Variables	Cases *N* (%)	Awareness		χ^2^	*p*-Value	Signed		χ^2^	*p*-Value
		Yes, *N* (%)	No, *N* (%)			Yes, *N* (%)	No, *N* (%)		
Total	3871	2771 (71.58)	1100 (28.42)			1952 (50.43)	1919 (49.57)		
Gender				1.278	0.261			0.014	0.906
Male	1571 (40.58)	1109 (70.59)	462 (29.41)			794 (50.54)	777 (49.46)		
Female	2300 (59.42)	1662 (72.26)	638 (27.74)			1158 (50.35)	1142 (49.65)		
Age				89.543	<0.001			422.768	<0.001
16–34	830 (21.44)	529 (63.73)	301 (36.27)			253 (30.48)	577 (69.52)		
35–49	1040 (26.87)	693 (66.63)	347 (33.37)			401 (38.56)	639 (61.44)		
50–64	761 (19.66)	547 (71.88)	214 (28.12)			406 (53.35)	355 (46.65)		
≥65	1240 (32.03)	1002 (80.81)	238 (19.19)			892 (71.94)	348 (28.06)		
Education Level				32.837	<0.001			155.201	<0.001
Elementary school or lower	1487 (38.41)	1092 (73.44)	395 (26.56)			932 (62.68)	555 (37.32)		
Junior middle school	1033 (26.69)	698 (67.57)	335 (32.43)			475 (45.98)	558 (54.02)		
High school	592 (15.29)	395 (66.72)	197 (33.28)			253 (42.74)	339 (57.26)		
Junior college	327 (8.45)	241 (73.70)	86 (26.30)			130 (39.76)	197 (60.24)		
Bachelor’s degree or higher	432 (11.16)	345 (79.86)	87 (20.14)			162 (37.50)	270 (62.50)		
Marital Status				4.693	0.096			53.767	<0.001
Single	348 (8.99)	234 (67.24)	114 (32.76)			126 (36.21)	222 (63.79)		
Married	3249 (83.93)	2332 (71.78)	917 (28.22)			1646 (50.66)	1603 (49.34)		
Divorced or widowed	274 (7.08)	205 (74.82)	69 (25.18)			180 (65.69)	94 (34.31)		
Household registration				3.343	0.067			8.499	0.004
Local	3785 (97.78)	2717 (71.78)	1068 (28.22)			1922 (50.78)	1863 (49.22)		
Non-local	86 (2.22)	54 (62.79)	32 (37.21)			30 (34.88)	56 (65.12)		
Personal Income (per month)				11.347	0.010			79.661	<0.001
≤2000	1642 (42.42)	1193 (72.66)	449 (27.34)			958 (58.34)	684 (41.66)		
2001–5000	1645 (42.49)	1136 (69.06)	509 (30.94)			711 (43.22)	934 (56.78)		
5001–8000	427 (11.03)	320 (74.94)	107 (25.06)			197 (46.14)	230 (53.86)		
≥8001	157 (4.06)	122 (77.71)	35 (22.29)			86 (54.78)	71 (45.22)		
Chronic disease history				187.98	<0.001			514.872	<0.001
Yes	1516 (39.16)	1273 (83.97)	243 (16.03)			1109 (73.15)	407 (26.85)		
No	2355 (60.84)	1498 (63.61)	857 (36.39)			843 (35.80)	1512 (64.20)		

**Table 2 ijerph-16-03312-t002:** Multivariate logistic regression analysis of the influencing factors associated with FDCS awareness.

Variables	B	SE	Wald Chi-Square Value	*p*-Value	OR (95% CI)
Constant	−0.381	0.131			
**Age** 16–34 (Ref.)					
35–49	0.455	0.113	16.109	<0.001	1.576 (1.262, 1.969)
50–64	0.66	0.136	23.418	<0.001	1.935 (1.481, 2.528)
≥65	1.023	0.144	50.501	<0.001	2.781 (2.098, 3.688)
**Education Level** Elementary school or lower (Ref.)					
Junior middle school	0.264	0.102	6.716	0.010	1.302 (1.066, 1.590)
High school	0.479	0.127	14.295	<0.001	1.614 (1.259, 2.069)
Junior college	1.107	0.167	43.961	<0.001	3.025 (2.181, 4.196)
Bachelor’s degree or higher	1.531	0.165	86.327	<0.001	4.623 (3.347, 6.386)
**Chronic Disease History** No (Ref.)					
Yes	1.058	0.095	123.197	<0.001	2.880 (2.390, 3.472)

B: regression coefficient; SE: standard error; OR: odds ratio; CI: confidence interval.

**Table 3 ijerph-16-03312-t003:** Multivariate logistic regression analysis of the influencing factors associated with contracting with family doctors.

Variables	B	SE	Wald Chi-Square Value	*p*-Value	OR (95% CI)
Constant	−2.916	0.176			
**Age** 16–34 (Ref.)					
35–49	0.096	0.126	0.583	0.445	1.101 (0.860, 1.408)
50–64	0.388	0.151	6.569	0.01	1.474 (1.096, 1.982)
≥65	0.977	0.155	39.584	<0.001	2.656 (1.959, 3.601)
**Education Level** Elementary school or lower (Ref.)					
Junior middle school	−0.165	0.118	1.944	0.163	0.848 (0.673, 1.069)
High school	−0.172	0.143	1.448	0.229	0.842 (0.636, 1.115)
Junior college	−0.286	0.181	2.506	0.113	0.751 (0.527, 1.071)
Bachelor’s degree or higher	−0.534	0.174	9.377	0.002	0.586 (0.417, 0.825)
**Personal Income (per month)** ≤2000 (Ref.)					
2001–5000	−0.303	0.097	9.679	0.002	0.739 (0.610, 0.894)
5001–8000	−0.017	0.147	0.013	0.91	0.984 (0.737, 1.312)
≥8001	0.604	0.216	7.836	0.005	1.829 (1.198, 2.791)
**Chronic Disease History** No (Ref.)					
Yes	0.967	0.098	97.21	<0.001	2.629 (2.170, 3.186)
**Awareness of FDCS** No (Ref.)					
Yes	3.125	0.123	642.608	<0.001	22.753 (17.870, 28.971)

B: regression coefficient; SE: standard error; OR: odds ratio; CI: confidence interval.

**Table 4 ijerph-16-03312-t004:** Comparison of FDCS demands among different residents (*n* = 3871) *N* (%).

Variables	Health Consultation	Follow-Up of Chronic Patients	Long-Term Prescriptions of Chronic Patients	Rehabilitation Guidance	Appointment Referral	Regular Physical Examination	Increasing the Proportion of MI Reimbursements	TCM Health Care	Family Bed Service	Free Door-to-Door Service
**Total**	3276 (84.63)	2550 (65.87)	2173 (56.14)	2512 (64.89)	2424 (62.62)	3163 (81.71)	3099 (80.06)	2291 (59.18)	1522 (39.32)	2253 (58.20)
**Sex**										
Male	1314 (83.64)	1004 (63.91)	889 (56.59)	1031 (65.63)	992 (63.14)	1253 (79.76)	1241 (78.99)	920 (58.56)	628 (39.97)	892 (56.78)
Female	1962 (85.30)	1546 (67.22)	1284 (55.83)	1481 (64.39)	1432 (62.26)	1910 (83.04)	1858 (80.78)	1371 (59.61)	894 (38.87)	1361 (59.17)
χ^2^	1.985	4.547	0.220	0.626	0.311	6.742	1.870	0.424	0.478	2.201
*p*-Value	0.159	0.033	0.639	0.429	0.577	0.009	0.172	0.515	0.489	0.138
**Age**										
16–34	682 (82.17)	455 (54.82)	408 (49.16)	520 (62.65)	557 (67.11)	620 (74.70)	653 (78.67)	489 (58.92)	340 (40.96)	478 (57.59)
35–49	885 (85.10)	627 (60.29)	541 (52.02)	658 (63.27)	679 (65.29)	851 (81.83)	860 (82.69)	621 (59.71)	394 (37.88)	581 (55.87)
50–64	627 (82.39)	506 (66.49)	427 (56.11)	469 (61.63)	441 (57.95)	634 (83.31)	583 (76.61)	434 (57.03)	259 (30.04)	409 (53.75)
≥65	1082 (87.26)	962 (77.58)	797 (64.27)	865 (69.76)	747 (60.24)	1058 (85.32)	1003 (80.89)	747 (60.24)	529 (42.66)	785 (63.31)
χ^2^	13.554	135.278	56.930	19.476	20.393	39.445	11.717	2.180	16.552	21.956
*p*-Value	0.004	<0.001	<0.001	<0.001	<0.001	<0.001	0.008	0.536	0.001	<0.001
**Education Level**										
Elementary school or lower	1254 (84.33)	1082 (72.76)	856 (57.57)	958 (64.43)	824 (55.41)	1277 (85.88)	1195 (80.36)	813 (54.67)	556 (37.39)	887 (59.65)
Junior middle school	869 (84.12)	647 (62.63)	568 (54.99)	657 (63.60)	655 (63.41)	861 (83.35)	821 (79.48)	606 (58.66)	373 (36.11)	574 (55.57)
High school	506 (85.47)	366 (61.82)	335 (56.59)	383 (64.70)	415 (70.10)	469 (79.22)	482 (81.42)	380 (64.19)	245 (41.39)	341 (57.60)
Junior college	277 (84.71)	201 (61.47)	168 (51.38)	220 (67.28)	227 (69.42)	251 (76.76)	257 (78.59)	213 (65.14)	148 (45.26)	193 (59.02)
Bachelor’s degree or higher	370 (85.65)	254 (58.80)	246 (56.94)	294 (68.06)	303 (70.14)	305 (70.60)	344 (79.63)	279 (64.58)	200 (46.30)	258 (59.72)
χ^2^	0.975	52.997	4.962	3.623	64.312	62.623	1.481	28.788	21.492	4.820
*p*-Value	0.914	<0.001	0.291	0.459	<0.001	<0.001	0.830	<0.001	<0.001	0.306
**Marital Status**										
Single	272 (78.16)	187 (53.74)	166 (47.70)	209 (60.06)	215 (61.78)	244 (70.11)	253 (72.70)	191 (54.89)	142 (40.80)	197 (56.61)
Married	2766 (85.13)	2161 (66.51)	1831 (56.36)	2117 (65.16)	2053 (63.19)	2689 (82.76)	2617 (80.55)	1944 (59.83)	1275 (39.24)	1894 (58.29)
Divorced or widowed	238 (86.86)	202 (73.72)	176 (64.23)	186 (67.88)	156 (56.93)	230 (83.94)	229 (83.58)	156 (56.93)	105 (38.32)	162 (59.12)
χ^2^	12.879	30.907	17.415	4.748	4.337	34.635	14.410	3.804	0.444	0.470
*p*-Value	0.002	<0.001	<0.001	0.093	0.114	<0.001	0.001	0.149	0.809	0.790
**Household Registration**										
Local	3198 (84.49)	2504 (66.16)	2133 (56.35)	2460 (64.99)	2372 (62.67)	3095 (81.77)	3030 (80.05)	2245 (59.31)	1488 (39.31)	2202 (58.17)
Non-local	78 (90.70)	46 (53.49)	40 (46.51)	52 (60.47)	52 (60.47)	68 (79.07)	69 (80.23)	46 (53.49)	34 (39.53)	51 (59.30)
χ^2^	2.490	6.002	3.308	0.757	0.174	0.410	0.002	1.181	0.002	0.044
*p*-Value	0.115	0.014	0.069	0.384	0.676	0.522	0.967	0.277	0.967	0.834
**Personal Monthly Income**										
≤2000	1396 (85.02)	1156 (70.40)	920 (56.03)	1037 (63.15)	955 (58.16)	1396 (85.02)	1317 (80.21)	912 (55.54)	611 (37.21)	984 (59.92)
2001–5000	1381 (83.95)	1033 (62.80)	915 (55.62)	1092 (66.38)	1064 (67.68)	1343 (81.64)	1329 (80.79)	1020 (62.01)	666 (40.49)	945 (57.45)
5001–8000	367 (85.95)	257 (60.19)	250 (58.55)	279 (65.34)	295 (69.09)	317 (74.24)	344 (80.56)	264 (61.83)	177 (41.45)	235 (55.04)
≥8001	132 (84.08)	104 (66.24)	88 (56.05)	104 (66.24)	110 (70.06)	107 (68.15)	109 (69.43)	95 (60.51)	68 (43.31)	89 (56.69)
χ^2^	1.380	28.059	1.193	3.944	28.278	47.287	11.757	15.788	5.862	4.302
*p*-Value	0.710	<0.001	0.755	0.268	<0.001	<0.001	0.008	0.001	0.119	0.231
**Chronic Disease History**										
No	1930 (81.95)	1289 (54.73)	1123 (47.69)	1424 (60.47)	1476 (62.68)	1863 (79.11)	1868 (79.32)	1369 (58.13)	914 (38.81)	1315 (55.84)
Yes	1346 (88.79)	1261 (83.18)	1050 (69.26)	1088 (71.77)	948 (62.53)	1300 (85.75)	1231 (81.20)	922 (60.82)	608 (40.11)	938 (61.87)
χ^2^	33.103	331.953	174.355	51.699	0.008	27.240	2.042	2.755	0.648	13.807
*p*-Value	<0.001	<0.001	<0.001	<0.001	0.929	<0.001	0.153	0.097	0.421	<0.001
**Contract with Family Doctor**										
No	1525 (79.47)	1088 (56.70)	959 (49.97)	1163 (60.60)	1131 (58.94)	1469 (76.55)	1491 (77.70)	1073 (55.91)	723 (37.68)	1073 (55.91)
Yes	1751 (89.70)	1462 (74.90)	1214 (62.19)	1349 (69.11)	1293 (66.24)	1694 (86.78)	1608 (82.38)	1218 (62.40)	799 (40.93)	1180 (60.45)
χ^2^	77.919	142.607	58.672	30.718	22.047	67.797	13.277	16.836	4.301	8.185
*p*-Value	<0.001	<0.001	<0.001	<0.001	<0.001	<0.001	<0.001	<0.001	0.038	0.004

MI: medical insurance; TCM: Traditional Chinese Medicine.

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
