# Peer review of "Residents’ Awareness of Family Doctor Contract Services, Status of Contract with a Family Doctor, and Contract Service Needs in Zhejiang Province, China: A Cross-Sectional Study"

_ijerph, 2019, doi:10.3390/ijerph16183312_

Round 1

Reviewer 1 Report

I suggest English editing to improve. Useful study. Abstract needs to include background a bit more.

Author Response

Comments and Suggestions for Authors

I suggest English editing to improve. Useful study. Abstract needs to include background a bit more.

Response: Thank you for this comment! We have rewritten our manuscript with the help of MDPI English Editing.

In the abstract section, we added the following content as background: “In China, family doctor services originated in 2009. After two years, the Chinese government proposed to establish a family doctor contract system suitable for China's national conditions. Then in 2016, a multi-department jointly issued a important document, which further clarified the development goals of family doctor contract services in the next five years. Zhejiang Province has been exploring responsible doctor contract services since 2012, which was promoted throughout the province in 2015.” Please see page 1, lines 15-20.

Reviewer 2 Report

Please find my comments as follows:

In the Introduction section, it would be great if you can give a brief introduction of primary care system in Zhejiang, at least including the following information: number of primary care centers, number of family doctors, their education level and qualification, number of residents being covered by each primary care center.

For Multi-stage sampling, can you give your justification of selecting 110 households per community or village?

For data collection, please report the following information: number of interviewers per county, selection criteria of interviewers, details of interviewer training, how to approach the participants, where to conduct the survey, how long it takes for each interview.

The tables should be significantly edited.

For Table 1, please add the sociodemographic information of the whole population in Zhejiang province, which will make it clear the representativeness of your sampling participant.

I don’t think Table 2 is necessary.

For Table 3 and Table 4, it seems the authors only report the variables which are statistically significant. Please report the results of all variables included in the regression models,

In the Discussion, could you please add the following information: among participants with awareness of contract service, how many of them don’t sign the contract and what their sociodemographic characteristics are; among the participants without awareness of contract service, how many of them sign the contract and what their sociodemographic characteristics are; these information will help identify the barriers of expanding contract service in China.

The authors compared the Zhejiang results with Beijing, Shanghai, and Guangzhou. Do the research participants in these cities have the similar sociodemographic characteristics with Zhejiang study? If yes, the comparison is ok. If not, it should be careful to interpret the comparison results.

In the fourth paragraph, instead of just reporting people with different characteristics have different needs, please explain why people with different characteristics have different needs.

In Conclusion, the authors mentioned: “Therefore, in order to promote the development of FDCS, strong policy support and publicity and targeted services developed for different groups are necessary.” Could you please give some practical policy suggestions? The language describing statistical analysis and reporting statistical results should be edited by a statistician.

Author Response

In the Introduction section, it would be great if you can give a brief introduction of primary care system in Zhejiang, at least including the following information: number of primary care centers, number of family doctors, their education level and qualification, number of residents being covered by each primary care center.

Response: Thank you for this comment! From the statistical data in the National Primary Health Information System, by the end of 2018, there are 1484 primary health care institutions that carry out contracted services, including 493 community health service centers and 991 township health centers, covering 52.83 million permanent residents. The average primary medical institution services 35,596 residents. The number of qualified doctors in Zhejiang Province has reached 24,228, and there are 19,896 registered as GPs, of which 19,341 are GPs who provide contract services and 9603 are GPs who have intermediate professional titles or above. We have added this to the manuscript. Please see page 3, lines 8-13.

For Multi-stage sampling, can you give your justification of selecting 110 households per community or village?

Response: Thank you for this important comment! To ensure the demographic representation, the sample quantity was calculated using the formula  with the sample size n = 3585. Specifically, confidence level α = 0.05 (two-sides),

 = 1.96, contracted rate p = 30% (according to the second-stage goal in the document mentioned above), allowable error δ = 0.015. To facilitate the allocation of places, the integer 3600 was taken as our sample size. With no response rate controlled within 10%, the actual sample size was 3960 residents, i.e. the sample size in average for each community or natural village was 110. We have added this content to the manuscript. Please see page 3, lines 30-35.

For data collection, please report the following information: number of interviewers per county, selection criteria of interviewers, details of interviewer training, how to approach the participants, where to conduct the survey, how long it takes for each interview.

 Response: Thank you for your comment! 

In this survey, we selected four to five interviewers per county.

The criteria for interviewers were to speak local dialects, to be good at communicating with others, to have relevant work experience, and to be responsible.

The details of interviewer training including the purpose and significance of the survey, the content of the questionnaire, the skills of the survey, the coding of the questionnaire, and the data entry.

At the beginning of the investigation, the selected householders were notified in advance to concentrate on one place to conduct the investigation. For the absent person, the investigators directly conducted the household survey. If the selected household is not available, choose the nearby neighborhood to replace, but the replacement rate was strictly controlled within 5%.

The location of the survey was generally selected at the Resident Activity Center, Cultural Auditorium or Community Health Service Center.

It takes about 3-5 minutes for each interview.

We have revised the manuscript. Please see page 6, lines 13-22.

The tables should be significantly edited.

For Table 1, please add the sociodemographic information of the whole population in Zhejiang province, which will make it clear the representativeness of your sampling participant.

Response: Thank you for this comment! Based on your suggestion, we have consulted a lot of relevant literature. However, we just obtained a small amount of relevant information from the 2018 Zhejiang Provincial Statistics Yearbook. By the end of 2017, the total population of Zhejiang Province is 37,509,600, including 19,482,900 males and 18,026,700 females. The age distribution of the registered population is: <18 years 8,408,336 (16.96%); 18-34 years 10,389,354 (20.96%); 35-59 years 19,947,643 (40.24%); ≥60 years 10,830,952 (21.85%). The educational level of the employed persons in order is: Junior high school and below 3,809,800; Technical secondary school and high school 2,555,700; junior college 1,654,400; Undergraduate and above 2,525,100.

These demographic information of the whole population in Zhejiang cannot be filled into the table 1. We are so sorry that we can not complete this task.

Regarding the representativeness of the sample, we have mentioned in the last paragraph of the discussion. In China, most young people go out to work and the elderly stay at home, therefore, the sampling population we surveyed were mostly older and female, which may lead to the underrepresentation of the whole population in Zhejiang province.

I don’t think Table 2 is necessary.

Response: Thank you for your suggestion! We have deleted the table and revised the manuscript.

For Table 3 and Table 4, it seems the authors only report the variables which are statistically significant. Please report the results of all variables included in the regression models,

Response: Thank you for your comment! 

The data was further subjected to binary logistic regression using the forward stepwise selection (Likelihood Ratio) method and using residents awareness of FDCS as the dependent variable; the factors significantly associated with awareness of FDCS (age, education level, personal income and chronic disease history ) as independent variables. Table 2 is the final result after three-step operation of the model. Please see the screenshot of the results below.

The logistic regression model used the contracted status (contracted and noncontracted) as the dependent variable, the factors significantly associated with contracting with FDs (age, education level, marital status, household registration, personal income, and chronic disease history), and residents’ awareness of FDCS or not as independent variables. Table 3 is the final result after five-step operation of the model. Please see the screenshot of the results below.

In the Discussion, could you please add the following information: among participants with awareness of contract service, how many of them don’t sign the contract and what their sociodemographic characteristics are; among the participants without awareness of contract service, how many of them sign the contract and what their sociodemographic characteristics are; these information will help identify the barriers of expanding contract service in China.

Response: Thank you for this important comment!  There were 913 participants who knew of family doctor contract services but did not sign up for them. Most of these were young people who had a high level of education, were married, were locals, had a moderate personal income level, and had no chronic disease history. Specifically, among such residents, 563 (61.66%) were female, 595 (65.17%) were younger than 50 years, 301 (32.97%) had an education level of junior college or higher, 757 (82.91%) were married, 888 (97.26%) were local residents, 457 (50.05%) had a personal monthly income of CN¥2001–5000, and 705 (77.21%) had no chronic disease history. For this group of people, we should first understand their needs and then provide corresponding services to meet their needs to attract them to sign up. Also, 94 of those who were not knowledgeable about FDCS had signed a contract. Most of them were elderly who had low levels of education, were married, were locals, had a low monthly income, and no chronic disease history. Specifically, 59 (62.77%) were female, 67 (71.28%) were older than 50 years, 59 (62.77%) had an education level of elementary school or lower, 71 (75.53%) were married, 93 (98.94%) were local residents, 56(59.57%) had a personal monthly income of ≤2000, and 50 (53.19%) had no chronic disease history. For such residents, it is necessary to strengthen publicity and encourage them to participate in contracted services.

We have added this content to the manuscript. Please see page 15, lines 7-21.

The authors compared the Zhejiang results with Beijing, Shanghai, and Guangzhou. Do the research participants in these cities have the similar sociodemographic characteristics with Zhejiang study? If yes, the comparison is ok. If not, it should be careful to interpret the comparison results.

Response: Thank you for you suggestion! Actually, we did not consider whether  the sociodemographic information of these provinces are consistent before comparison. But as we know, the provinces with the highest economic development levels include Beijing, Shanghai, Guangdong, Zhejiang and Jiangsu in China. Therefore, according to the level of economic development, they are comparable.

On the other hand, because Beijing, Shanghai and Guangdong are the first regions to implement family doctor contract services. We believe that if we want to make a big progress, the best way is to find a gap with them.

In the fourth paragraph, instead of just reporting people with different characteristics have different needs, please explain why people with different characteristics have different needs.

Response: Thank you for you suggestion! Because each participant's physical condition (gender, age, chronic disease history), education level, and personal monthly income level are different, the demand for contracted services is different. We have revised the manuscript. Please see page 15, lines 25-27.

In Conclusion, the authors mentioned: “Therefore, in order to promote the development of FDCS, strong policy support and publicity and targeted services developed for different groups are necessary.” Could you please give some practical policy suggestions? The language describing statistical analysis and reporting statistical results should be edited by a statistician. 

Response: Thank you for your suggestion! Based on the results of this survey, in order to promote the development of FDCS, the government should provide greater policy support, increase publicity, expand service packages, and provide more attractive service projects. Residents can choose a family doctor with whom to sign a contract. When signing the contract, residents can independently choose the items they need. At the same time, family doctors should provide better services to enhance residents’ sense of fulfillment and satisfaction. We have added this content to the manuscript. Please see page 16, lines 28-33.

Reviewer 3 Report

This is an interesting article of importance in the reform of the reform of the medical and health care system in China. The article is well written but I have outlined some concerns that the authors need to address.

Abstract

Replace evidence-based suggestions with evidence based recommendations

The first sentence is too long and should be broken into two sentences

Under design, please do not start a sentence with a numerical value

The authors stated that a self-designed questionnaire was conducted with them – please replace with “a survey using a self-designed questionnaire was used to collect data-----,

Please clarify what you mean by responsive questionnaires

Be consistent in using the past and present tense in the presentation of the results

Introduction

The introduction is well written and the justification for the study is well articulated but the authors should address the following:

Page 2 line 45, please clarify what you mean by on-the-spot investigations.

Page 2 line 49, replace research objects with research participant or study population;

Page 2 line 50, replace Solve the following problems with answer the following questions;

Page 3 line 2, replace problems will provide suggestions with provide recommendations

Methodology

The sample included participants aged 16 years and above. Please clarify the reason for the inclusion of people as young as 16 years in the study, what is their involvement in the contracting with FDs?

Data collection

Page 5 line 16-17, it is stated that a small gift was sent to each respondents to improve the residents’ cooperation and responding rate. What was the gift? What are the ethics of this kind of practice in the country?

Page 5 line 18-19, in case of the household absence, the selected household could be replaced by the nearby neighborhood-please rephrase to read better.

Results

Page 5 line 39-40, it is stated that 1571 (40.58%) were male, 2300 (49.42%) were female, the distribution does not add up to 100%.

Page 5 line 41, the age group are 18-35 years, 35-50 years, 50-65 years and 65 and above. There is overlapping of the age in all the groups. If someone is 50 years old, how is it that the individual will be in two groups, how was age collected? The authors need to correct the age categories and reanalyse the data for this variable.

Page 7 line 19-21, please rephrase the sentence for clarity “the logistic regression model uses the contracted status (contracted and non-contracted) as the dependent variable, uses the factors significantly associated with contracted with FD which fore mentioned and residents awareness of FDCS or not as independent variables”.

Discussion

The contract rate in Zhejiang province of 50.43% is compared to rates of different provinces than the awareness rate of 7i.58% which was compared to Beijing, Guangdong and Shanghai provinces. Please clarify the discrepancy in the choice of comparison for the two outcomes of the study.

Page 12 line 5, please delete reference to table5 in the discussion

Page 12 line 22, please delete burden et al, and completed the sentence

Page 13 line 4, please rephrase “at the same time, do residents know that FDCS affect 5 residents’ contract with family doctors”.

General comments

There are grammar and tense errors throughout the document that need to be addressed.

Author Response

Abstract

Replace evidence-based suggestions with evidence based recommendations

Response: Thank you for your suggestion! We have update the manuscript. Please see page 1, line 24.

The first sentence is too long and should be broken into two sentences

Response: Thank you for this important comment! According to your suggestion, I have divided the first sentence into two sentences. Please see page 1, line 20-24.

Under design, please do not start a sentence with a numerical value

Response: Thank you for your suggestion! We have edited the sentence. Please see page 1, lines 25-26.

The authors stated that a self-designed questionnaire was conducted with them – please replace with “a survey using a self-designed questionnaire was used to collect data-----,

Response: Thank you for your suggestion! We have update the manuscript. Please see page 1, line 26.

Please clarify what you mean by responsive questionnaires

Response: Thank you. In our manuscript, “responsive questionnaires” means these questionnaires were valid. We have replace responsive questionnaires with valid questionnaires. Please see page 1, line 29.

Be consistent in using the past and present tense in the presentation of the results

Response: Thank you for this comment! We have rewritten our manuscript with the help of MDPI English Editing.

Introduction

The introduction is well written and the justification for the study is well articulated but the authors should address the following:

Page 2 line 45, please clarify what you mean by on-the-spot investigations.

Response: Thank you for this comment! On-the-spot investigation means field investigation.

Page 2 line 49, replace research objects with research participant or study population;

Response: Thank you for your suggestion! We have revised the manuscript. Please see page 3, line 19.

Page 2 line 50, replace Solve the following problems with answer the following questions;

Response: Thank you for your suggestion! We have update the manuscript. Please see page 3, line 20.

Page 3 line 2, replace problems will provide suggestions with provide recommendations

Response: Thank you for your suggestion! We have update the manuscript. Please see page 3, line 24.

Methodology

The sample included participants aged 16 years and above. Please clarify the reason for the inclusion of people as young as 16 years in the study, what is their involvement in the contracting with FDs?

Response: Thank you for this important comment! At present, in China, the contracted services of family doctors have given priority to covering key populations, including the elderly, pregnant women, children and the disabled, as well as patients with chronic diseases such as hypertension, diabetes and tuberculosis, patients with severe mental disorders, special family members with family planning, rural people with difficulties. On this basis, strive to expand the contract service to the entire population. 

We defined the eligibility criteria of participants was 16 years and above. One is that people younger than 16 are not the primary target of family doctors’ services. The secondly, considering that the cognitive ability of people younger than 16 years old is limited, they may not be able to complete the questionnaire independently.

Data collection

Page 5 line 16-17, it is stated that a small gift was sent to each respondents to improve the residents’ cooperation and responding rate. What was the gift? What are the ethics of this kind of practice in the country?

Response: Thank you for your questions! Gifts for investigation include towels, toothpaste, toothbrushes, rags, etc. In the actual survey process, in order to improve residents' cooperation, residents can get a small gift as a reward when they complete a questionnaire. Of course, not all field surveys like this have gifts as rewards. It depends on various factors such as the project's funds, survey methods, and respondents.

There is a strict ethical review system in China. However, for field investigations, ethical review may be waived because personal information such as name, ID card, and biological samples such as blood and urine are not involved.

Page 5 line 18-19, in case of the household absence, the selected household could be replaced by the nearby neighborhood-please rephrase to read better.

Response: Thank you for this comment! We have changed this sentence. Please see page 6, lines 21-22.

Results

Page 5 line 39-40, it is stated that 1571 (40.58%) were male, 2300 (49.42%) were female, the distribution does not add up to 100%.

Response: Thank you for noting this. An error occurred here due to my negligence. Here, 2300(49.42%) should be 2300(59.42%). We have revised this in our manuscript. Please see page 7, lines 1.

Page 5 line 41, the age group are 18-35 years, 35-50 years, 50-65 years and 65 and above. There is overlapping of the age in all the groups. If someone is 50 years old, how is it that the individual will be in two groups, how was age collected? The authors need to correct the age categories and reanalyse the data for this variable.

Response: Thank you for this important comment! I am sorry to have troubled you for my statement is not clear. Actually, we use “~” instead of  “-” in our manuscript. 16~35 years indication includes 16 years old but not 35 years old. In order to make the statement clearer, we have revised the manuscript.

Page 7 line 19-21, please rephrase the sentence for clarity “the logistic regression model uses the contracted status (contracted and non-contracted) as the dependent variable, uses the factors significantly associated with contracted with FD which fore mentioned and residents awareness of FDCS or not as independent variables”.

Response: Thank you for your suggestion! We have revised the manuscript. Please see page 9, lines 7-10.

Discussion

The contract rate in Zhejiang province of 50.43% is compared to rates of different provinces than the awareness rate of 71.58% which was compared to Beijing, Guangdong and Shanghai provinces. Please clarify the discrepancy in the choice of comparison for the two outcomes of the study.

Response: Thank you for your comment!  We are not sure we understand exactly what your problem is.

The first one is why we compared the Zhejiang results with Beijing, Shanghai, and Guangzhou. There are two reasons for this. Firstly, as we know, the provinces with the highest economic development levels include Beijing, Shanghai, Guangdong, Zhejiang and Jiangsu at present in China. Therefore, according to the level of economic development, they are comparable. Secondly, because Beijing, Shanghai and Guangdong are the first regions to implement FDCS. We believe that if we want to make a big progress, the best way is to find a gap with them.

The second one is what is the reason for the different. As we mentioned in the discussion section, the signing service for FDs in Zhejiang Province starting late was one of the main reasons. In addition, it was related to the different levels of economic development, geographical location, survey population and policy implementation in these provinces.

If we did not answer your question exactly, please don't hesitate to notify me in time. Thank you so much.

Page 12 line 5, please delete reference to table5 in the discussion

Response: Thank you for your suggestion! We have revised the manuscript. Please see page 15, line 24.

Page 12 line 22, please delete burden et al, and completed the sentence

Response: Thank you for your suggestion! We have revised the manuscript. Please see page 15, lines 42-43.

Page 13 line 4, please rephrase “at the same time, do residents know that FDCS affect 5 residents’ contract with family doctors”.

Response: Thank you for your suggestion! We have revised the manuscript. Please see page 16, lines 25-26.

General comments

There are grammar and tense errors throughout the document that need to be addressed.

Response: Thank you for this comment! We have rewritten our manuscript with the help of MDPI English Editing.
